# SHORTCIRCUIT: ALPHAZERO-DRIVEN GENERATIVE CIRCUIT DESIGN

## ABSTRACT

Chip design relies heavily on generating Boolean circuits, such as AND-Inverter Graphs (AIGs), from functional descriptions like truth tables. This generation operation is a key process in logic synthesis, a primary chip design stage. While recent advances in deep learning have aimed to accelerate circuit design, these efforts have mostly focused on tasks other than synthesis, and traditional heuristic methods have plateaued by primarily optimizing small 4-inputs cuts. In this paper, we introduce ShortCircuit, a novel transformer-based architecture that leverages the structural properties of AIGs and performs efficient space exploration. Unlike prior approaches that attempt end-to-end generation of logic circuits using deep networks, ShortCircuit employs a two-phase process combining supervised learning with reinforcement learning to enhance generalization to unseen truth tables. We also propose an AlphaZero variant to handle the doubly exponential state space and the reward sparsity, enabling the discovery of near-optimal designs. To evaluate the generative performance of our model, we extract 500 8-input truth tables from a set of 20 real-world circuits. ShortCircuit guarantees the correctness of the produced AIGs, and outperforms the state-of-the-art logic synthesis tool `ABC`, by $18.62\%$ with respect to circuit size, while its greedy rollout is $31\times$ faster.

## 1 INTRODUCTION

The rapid rise of AI has driven computational demands beyond current hardware capabilities, creating a major bottleneck. Chip design is key to advancing next-generation computing, but traditional methods struggle to keep pace, highlighting the need for faster and more innovative approaches. At its core, a chip is the physical embodiment of a Boolean function, transforming binary inputs into desired outputs. Creating these embodiments is facilitated by logic synthesis, a crucial step in chip design that converts functional descriptions into graphs connecting logic gates. The logic synthesis process must balance power, performance, and area, posing a complex optimization challenge. In this paper, we investigate the use of Machine Learning (ML) to generate optimized digital circuits directly from Boolean logic specifications, offering a fresh perspective on the chip design process.

Truth tables fully specify Boolean functions by listing outputs for all input combinations. Consequently, we use truth tables as the input Boolean logic description for our problem. Our method outputs directed acyclic graphs (DAGs) as AND-Inverter Graphs (AIGs), a standard logic structure in Electronic Design Automation (EDA) Mishchenko and Brayton (2006); Wolf et al. (2013). AIGs only use 2-input AND gates and inverted or normal edges, offering a simple, scalable, and widely adopted representation that makes them an ideal choice for our ML-based circuit generation approach.

Traditional logic optimization methods optimize large AIGs by forming small cuts, finding smaller equivalent representations, and replacing the original subgraphs. The most effective method, `rewrite`, forms 4-input cuts and matches their truth tables to a precomputed database for replacements (Mishchenko et al., 2006). While industrial approaches have explored scaling to 8-input cuts, the double exponential growth of the truth table space limits the coverage of such databases (Amarú et al., 2017). Moreover, the structural rigidity of AIGs and the complexity of the problem result in most cuts failing optimization, limiting the effectiveness of these methods (Tsaras et al., 2025). Recent works aim to speed up chip design by applying ML across EDA stages (Huang et al., 2021; Gubbi et al., 2022), including placement (Ward et al., 2012), routing (Alawieh et al., 2020), and logic synthesis (Tu et al., 2024). Rather than directly tackling the graph generation problem, most ML

methods for logic synthesis focus on the optimization of synthesis recipes, which are sequences of operators acting on a logic graph to modify its structure while preserving the associated Boolean function. More recently, deep-generative methods emerged aiming to generate logic graphs rather than operator sequences d'Ascoli et al. (2024); Li et al. (2025); Dong et al. (2023). The generative approach offers higher potential as it offers more flexibility than working with a fixed set of operators; however, this approach requires exploring a much larger space, due to the double exponential growth of the search space with the number of Boolean function inputs.

Such a vast search space renders traditional ML methods ineffective for optimal AIG generation. However, recent advances have demonstrated that tailored model architectures and exploration-exploitation-aware training protocols can achieve remarkable performance even in tasks involving such large spaces. Indeed, the success of methods like AlphaGO Silver et al. (2016), AlphaZero Silver et al. (2017), AlphaFold Jumper et al. (2021) and even the emergence of large language models Devlin et al. (2019); Kaplan et al. (2020) relied on the development of custom model architectures capturing structural properties of board games, proteins, or language. Moreover, the training of these models either leverages naturally abundant data or employs specific data-augmentation and exploration-exploitation strategies to improve their performance.

In this paper, we propose ShortCircuit, a new transformer-based architecture structurally adapted for generating AIGs. Our transformer takes logic nodes represented as truth tables as input, and each forward pass predicts the next AND node to create in order to realize a target truth table. We further utilize an AlphaZero policy to navigate the large state space and discover more compact designs. We make the following contributions: **i)** we formally define the challenging problem of generating AIGs from target truth tables, characterized by a doubly exponential state space and a quadratically growing action space. **ii)** We introduce ShortCircuit, a novel AIG-aware architecture, enabling effective exploration of this vast search space that guarantees the correctness of the produced AIG. **iii)** We develop a two-stage training scheme combining supervised learning and reinforcement learning to improve generalization and scalability. Finally, **iv)** we empirically demonstrate the effectiveness of ShortCircuit by producing circuits $18.62\%$ smaller compared to the state-of-the-art logic synthesis tool `ABC` (Mishchenko et al., 2007), while achieving a $31\times$ speedup in its greedy variant, showcasing the potential of ML methods to revitalize the field of logic synthesis with a fresh perspective.

We present background and related work in Section 2, followed by the problem formulation in Section 3. We then describe our model architecture in Section 4 and our tailored training procedure in Section 5. We finally provide the empirical evaluation of ShortCircuit in Section 6.

## 2 BACKGROUND

A digital circuit cascades logic gates to realize a Boolean function $f : \{0,1\}^n \rightarrow \{0,1\}^m$, mapping $n$-bit inputs to $m$-bit outputs. An And-Inverter Graph (AIG) is a Directed Acyclic Graph (DAG) that is commonly used to represent such functions at the early stage of the chip design process.

### 2.1 AND-INVERTER GRAPHS

An AIG is composed of three types of nodes: (1) primary inputs, which we also refer to as inputs, (2) primary outputs, which we call the outputs, and (3) 2-input AND-nodes representing the logic gate AND. In this work, we focus on the generation of single-output AIGs that represent Boolean functions of the form $f : \{0,1\}^n \rightarrow \{0,1\}$ as they play an important role in logic synthesis. Fig. 1 illustrates the structure of an AIG with $n = 3$ inputs, where $\{I_k\}_{1 \leq k \leq 3}$ represent input nodes, $\wedge_4, \wedge_5$ are AND gates, and $O$ is the output. Edge orientation indicates the direction of the Boolean signal propagation from one node (called fanin) to another (called fanout). Moreover, the two types of edges, plain and dashed, in Fig. 1 indicate that the Boolean signal can be inverted when going from a fanin to a fanout. The primary output is always connected to a single AND node with a direct or an inverter link. As AIGs only contain AND operations and Boolean inversions, we can map them to a canonical form (CNF). For instance, the CNF, naturally derived from its topology, of the AIG in Fig. 1 is $O = \neg(\neg(I_1 \wedge I_2) \wedge I_3)$, and we can conversely easily go from a CNF to an AIG. Moreover, applying equivalence-preserving operations to the CNF produces new CNFs that still encode the same Boolean function. Similarly, topologically distinct AIGs can realize the same function, and to compare the quality of two AIGs, a primary criterion is to compare their sizes, measured by the

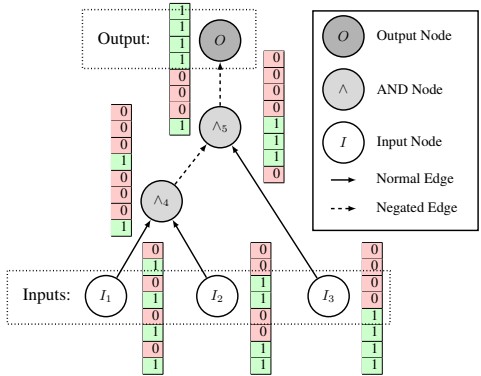

Figure 1: Representation of an AIG, showing the truth table associated to each node.

Table 1: Truth table of each node appearing in the AIG from Fig. 1

| $I_3$ | $I_2$ | $I_1$ | $I_1$ | $I_2$ | $I_3$ | $\wedge_4$ | $\wedge_5$ | O |
|---|---|---|---|---|---|---|---|---|
| 0 | 0 | 0 | 0 | 0 | 0 | 0 | 0 | 1 |
| 0 | 0 | 1 | 1 | 0 | 0 | 0 | 0 | 1 |
| 0 | 1 | 0 | 0 | 1 | 0 | 0 | 0 | 1 |
| 0 | 1 | 1 | 1 | 1 | 0 | 1 | 0 | 1 |
| 1 | 0 | 0 | 0 | 0 | 1 | 0 | 1 | 0 |
| 1 | 0 | 1 | 1 | 0 | 1 | 0 | 1 | 0 |
| 1 | 1 | 0 | 0 | 1 | 1 | 0 | 1 | 0 |
| 1 | 1 | 1 | 1 | 1 | 1 | 1 | 0 | 1 |

number of gates they contain (Mishchenko and Brayton, 2006). Smaller AIGs are generally preferred as they simplify subsequent tasks such as placement and routing, and lead to more efficient circuits.

## 2.2 TRUTH TABLES

As for any other logical graphs, we can capture the behavior of an AIG by propagating Boolean values from its primary inputs to its primary outputs and applying the logical operations encountered on the directed paths. Considering again the exemplar AIG from Fig. 1, if $I_1 = 1$, $I_2 = 1$, and $I_3 = 0$, propagating the Boolean signals we can verify that the AIG output at $O$ is 1. By enumerating all possible input combinations and recording the corresponding output values for each gate, we can build the AIG's full truth table, as shown in Table 1. Each row corresponds to the values of the AIG nodes for a specific set of entries $(I_1, I_2, I_3) = (i_1, i_2, i_3) \in \{0, 1\}^3$ displayed on the left part of the table. We can extract from this representation a binary vector of size $2^n$ for each AIG node. For instance, the "truth table" vector representation of node $\wedge_4$ is $(0\ 0\ 0\ 1\ 0\ 0\ 0\ 1)^\top$, and the primary output one is $(1\ 1\ 1\ 1\ 0\ 0\ 0\ 1)^\top$. After discussing related works in more detail, we will explain in the next section how our method leverages this rich vector representation for AIG generation.

## 2.3 RELATED WORKS

We discuss heuristics for AIG generation and ML methods for logic operator sequence optimization in Appendix B, and focus here on the deep learning approaches tailored to logic graph generation.

**Learning to Generate One Circuit at a Time** A first approach to generate Boolean networks with deep neural networks consists of substituting the gates and wires of a logic circuit by learnable nodes and connections to form a neural network (Belcak and Wattenhofer, 2022; Zimmer et al., 2023; Hillier et al., 2023). The network parameters are learnt by minimizing the error made by forward passes compared to the target binary output. On the one hand, this method allows to cope with larger number of primary inputs as, instead of learning an entire family of logic graphs (e.g., the 8-input AIGs), it is specialized on one particular target truth table. On the other hand, it requires to train a new neural network for each target truth table, representing a significant runtime bottleneck. Therefore, several works inspired by the development of foundational generative models have followed another direction, consisting in learning the synthesis process itself with a deep neural network.

**Learning Circuit Synthesis using Deep Learning** While Roy et al. (2021) use a CNN backbone to generate prefix circuits by adding or deleting nodes in an $N \times N$ grid, Li et al. (2024) employ an auto-regressive diffusion model to generate DAG for high-level synthesis stage , and Dong et al. (2023) design a two-level GNN architecture to synthesize analog circuits that work with non-binary signals. Closer to our work are Boolformer (d'Ascoli et al., 2024) and Circuit Transformer (CT) (Li et al., 2025), both tackling digital network synthesis with an auto-regressive transformer-based architecture. They train their policies via supervised learning by predicting the next element of a logic graph given a target truth table, and do inference through beam-search or MCTS simulations. Contrary to our

work, Boolformer and CT use a symbolic representation of the Boolean formula associated to a logic graph encoded via depth-first search. Therefore, instead of representing already built nodes by their truth tables and predicting the next node to add to the graph, they tokenize the symbols and let their model generate the next symbol of the logic formula (a logical operation, an input or an output), which requires more forward passes than for our method to produce the same AIG. Besides, given a target truth table $T_\star$, Boolformer directly takes as input the boolean representation of $T_\star$ as we do, while CT takes a (non-optimal) AIG realizing $T_\star$ and generates an improved logic network.

## 3 PROBLEM DEFINITION

Truth tables encompass the results for all possible input values, but do not provide sufficient structural information to derive an AIG representing that mapping. Heuristics that generate a Boolean expression, or an AIG, from a truth table lead to exponentially large solutions needing further refinement. Consequently, solving this AIG generation problem would significantly impact digital circuit design.

**Problem Definition:** Given a target truth table $T_\star \in \{0,1\}^{2^n}$, construct an AIG with the minimum number of nodes, such that its output node $O$ has a truth table $T_O$ matching $T_\star$.

Note that the size of a truth table associated with an $n$-input AIG is $2^n$, thus, the set containing all truth tables of size $2^n$ has a cardinality of $2^{2^n}$, which is on the order of $10^{77}$ for the space of truth tables that an 8-input AIG can represent. Exploring such a large space efficiently requires developing specific models and training techniques. These must exploit the structural properties of the problem while managing the exploration-exploitation trade-off inherent in such scenarios.

### 3.1 STATE REPRESENTATION & NOTATIONS

We formally define an AIG with $n$ inputs as a graph $\mathcal{G} = (V, E)$, where $V$ and $E$ represent the node and edge sets. To capture the dynamic nature of AIG construction, we introduce a temporal parameter, $t$, which simultaneously represents the current time step and the number of AND-nodes in the graph, denoted as $\mathcal{G}_t = (V_t, E_t)$. This allows us to model the evolution of the AIG over time, with the graph growing as new AND-nodes are added. We assign a unique integer ID to each node in the graph, regardless of its type; thus, at time $t$ the node set is $V_t = \{I_1, \ldots, I_n, \wedge_{n+1}, \ldots, \wedge_{n+t}\}$, or with a node-type agnostic notation, $V_t = \{v_1, \ldots, v_{n+t}\}$. Following this notation, the graph $\mathcal{G}_0$ represents an AIG containing only input nodes, i.e., with $V_0 = \{I_1, I_2, ..., I_n\}$.

In our generative process, we encode the state corresponding to AIG $\mathcal{G}_t$ as a 3-tuple $s_t = (\mathcal{T}_t, T_\star, \mathcal{A}_t)$. Here, $\mathcal{T}_t = \{T_1, T_2, ..., T_{n+t}\}$ is the set of truth tables associated with the current nodes, $T_\star$ is the target truth table, and $\mathcal{A}_t = \{a_{n+1}, a_{n+2}, ..., a_{n+t}\}$ is the ordered set of actions performed so far. Each action generates a new AND-node by connecting two existing nodes in one of four possible configurations: $(v_i, v_j)$, $(v_i, \neg v_j)$, $(\neg v_i, v_j)$, and $(\neg v_i, \neg v_j)$. Our goal is to perform a series of $N$ actions, transforming $\mathcal{G}_0$ into a terminal AIG $\mathcal{G}_N$ such that the truth table of the last generated AND-node, $T_{n+N}$, matches or is the negation of the target truth table $T_\star$. Note that our environment is stateful, as its history influences future decisions. Furthermore, our environment poses an additional challenge due to its action space expanding quadratically with each new node we add.

## 4 MODEL ARCHITECTURE

We propose an iterative approach to AIG construction, where we gradually build the circuit, starting with $\mathcal{G}_0$ and letting our model decide at each step which AND-node to add, aiming at realizing a given target truth table $T_\star$. To generate the next gate, the model takes the set of existing nodes as input, and it outputs a probability distribution over the set of AND-nodes that can be built by combining any pair of already existing nodes, taking edge types into account. Formally, let $|V_t|$ denote the number of nodes in the current state of the graph, then the action space is a $4 \times |V_t| \times |V_t|$ tensor, where each $|V_t| \times |V_t|$ slice corresponds to a connection type. Specifically, the cell $(i, j)$ in a given slice indicates the probability of connecting node $v_i$ with $v_j$, and the slice index $\epsilon \in \{1, 2, 3, 4\}$, corresponds to a combination of specific edge types: $(v_i, v_j)$, $(\neg v_i, v_j)$, $(v_i, \neg v_j)$, or $(\neg v_i, \neg v_j)$. Therefore, we can sample a triplet $(\epsilon, i, j)$ following the distribution given by this $4 \times |V_t| \times |V_t|$ tensor and add the corresponding node to the graph. The process ends when the truth table of the sampled node matches either the target one $T_\star$ or its full negation $\neg T_\star$, or after reaching a maximum number of steps $N_{\max}$.

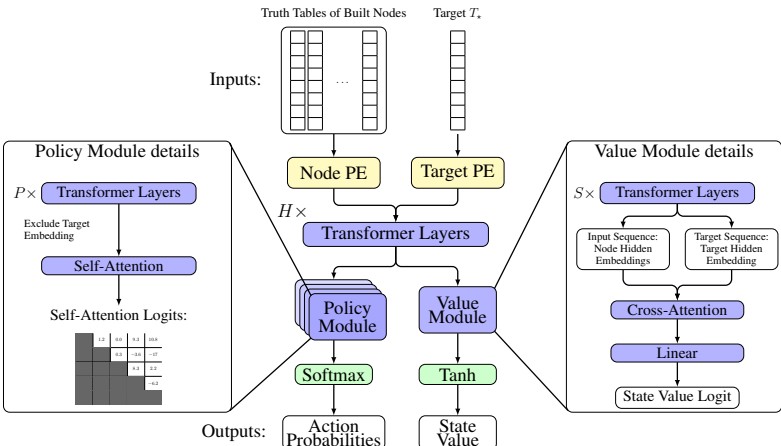

Figure 2: ShortCircuit model takes as inputs a target truth table $T_\star$ and the truth tables of the already built nodes. It first appends a type-dependent positional encoding before going through several transformer layers. Then, the model is split into two heads respectively outputting a probability distribution over the next possible actions (policy module on the left), and a value reflecting the quality of the current inputs (value module on the right).

To effectively explore and prune the vast state space, our model comprises a policy and a value module to assess intermediate states and strategically get closer to the desired target. As shown on Fig. 2, our architecture consists of a shared core embedding the truth tables applying position encodings and $H$ transformer encoder layers. Then, the hidden embeddings are passed as input to the 4 stacked policy modules and to the value module. The policy modules combined with a softmax produce a distribution over next actions, and the value module ending with $\tanh$ predicts an expected reward in $[-1, 1]$.

| $\epsilon$ | Build $v_k$ from $(\epsilon, i, j)$ | | | | |
|---|---|---|---|---|---|
| 1 | | $v_i$ | $\wedge$ | | $v_j$ |
| 2 | $\neg$ | $v_i$ | $\wedge$ | | $v_j$ |
| 3 | | $v_i$ | $\wedge$ | $\neg$ | $v_j$ |
| 4 | $\neg$ | $v_i$ | $\wedge$ | $\neg$ | $v_j$ |

Table 2: Edges for each $\epsilon$.

**Positional Encoding (PE)** In transformers, positional encodings (PE) help capture the sequential relations in the inputs. In our setting, the AIG structure is already implicitly reflected in the truth tables. Furthermore, self-attention should treat every node equally, as any two nodes may be combined to form a new node. However, to enable the model to distinguish between built nodes and the target truth table, we introduce two learnable positional encoders, referred to as "Node PE" (see Fig. 2).

**Policy Module** Transformers are the state-of-the-art architecture to handle sequential data. These models particularly shine when trained to predict the next token in a sequence by outputting a fixed-size tensor representing a sampling probability over a token glossary. In our case though, the set of nodes that we can build grows at each step, as more pairs of nodes can be combined to produce the next node. Inspired by NLP tasks, for which attention scores among related tokens are high, we use attention to guide next node generation. Thus, we directly use the final self-attention map of four parallel policy modules to get the probability to connect any two nodes with specific edge types. Each policy module has $P$ transformer encoder layers and outputs a final self-attention layer. In the last self-attention layer, we exclude the entry corresponding to the target node and returns the self-attention scores based on the existing AIG nodes embeddings. We aggregate the scores from the four policy modules and mask the ones associated to already built nodes and their negate versions. We finally apply a softmax to the remaining scores to produce a single probability distribution.

**Value Module** The value module consists of $S$ transformer encoder layers. It assess how favorable a state is and prevents expanding unpromising states. As the quality of a state not only depends on the nodes that are present in the graph at that stage, but also on the target truth table that should be realized, the value module also uses the two learnable type-based positional encoders introduced above. After performing the embedding, we compute the cross-attention between the graph nodes and the target truth table, which yields a new vector representation of the target. Finally, we feed this vector to a linear layer producing a single value, which should reflect the quality of the current state.

## 5 TRAINING SHORTCIRCUIT

The sparse nature of the problem makes it practically impossible to discover functionally correct graphs when exploring the double exponential state space uniformly. This challenge necessitates a more effective training approach for our ShortCircuit. To address this, we propose a two-stage training regimen consisting of a supervised pre-training stage to initialize the policy module, followed by an AlphaZero-style fine-tuning phase to improve the policy module and train the value module. Although pre-training the policy module provides a good prior to predict the most useful next actions, simply following it does not guarantee that an AIG matching $T_\star$ will be constructed due to the inherent difficulty of the problem. This limitation highlights the importance of the fine-tuning stage, which aims to refine the policy module and leverage the value module for improved performance.

### 5.1 PRE-TRAINING

Inspired from NLP, we pre-train our model on the next-action prediction task using single-output AIGs. As no large public corpus of (truth table, AIG) pairs exists, we curate our own dataset from open-source digital circuit collections to pre-train ShortCircuit to regenerate AIGs from their targets.

**Data Extraction** To generate an AIG dataset with the desired input and output sizes, we utilize the EPFL benchmarks (Amarú et al., 2015), which contain a collection of 20 real circuits realizing arithmetic and control functions. On average, the arithmetic and control circuits have 175 inputs and 137 outputs with 22520 AND-nodes, as detailed on Table 4 and 5. Since these circuits have more inputs than the AIGs we aim to generate, we extract subgraphs, or *cuts*, from them. A cut refers to a connected subset of nodes in the AIG that divides the graph into two disjoint parts. The root of a cut is the node to which all directed paths within the cut converge to and a *leaf node* is a node in the cut that have at least one fanin outside of the cut. By design, a cut forms a single-output AIG with a number of inputs corresponding to the number of leaves. We defer to Appendix C.2 the full description of the cut extraction method we develop to build a dataset of single-output AIGs.

**Data Preparation** Since our policy should predict the next action, i.e. the next node to add to a partial AIG, we need to convert the AIGs we load from our training dataset into a sequence of ground-truth actions. As different series of actions can lead to the same graph with $N$ nodes, we first sort the nodes of the training AIG we load into a topological order $\{I_1, \ldots, I_n, \wedge_{n+1}, \ldots, \wedge_N\}$ (or $\{v_1, \ldots, v_N\}$ with node-type agnostic notation), where $\wedge_N$ is connected to the output $O$. We also convert the AIG nodes into truth tables, as described in section 2, and use the truth table of $O$ as the target $T_\star$. From the topological sequence of nodes, we build the sequence of actions that our policy should learn to perform when its goal is to generate $T_\star$. As mentioned in section 4, creating node $v_k = (\neg)v_i \wedge (\neg)v_j$, with $1 \le i < j < k$, corresponds to action $a_k = (\epsilon, i, j)$, whose first component $\epsilon \in \{1, 2, 3, 4\}$, indicates the types of the edges connecting $v_k$ to its parents, as detailed in Table 2. This procedure leaves us with the sequence of actions $\mathcal{A} = \{a_{n+1}, a_{n+2}, ..., a_N\}$, starting with index $n+1$ since the $n$ primary inputs are given at the beginning of the AIG generation process.

To efficiently generate target action distributions, we aggregate all the actions into a sparse 3-dimensional tensor $\mathbf{A} = (\mathbf{A}_1, \mathbf{A}_2, \mathbf{A}_3, \mathbf{A}_4)$ where each element $\mathbf{A}_\epsilon$ is a $N \times N$ matrix representing the actions with connection type $\epsilon$. The value of the entry $(i, j)$ of $\mathbf{A}_\epsilon$ is set to 1 if $(\epsilon, i, j)$ belongs to $\mathcal{A}$ and to zero otherwise. Thus, if all the nodes up to $v_k$ are already built, considering each submatrix $\mathbf{A}_{\epsilon, 1:k, 1:k}$ taking the first $k$ rows and $k$ columns of $\mathbf{A}_\epsilon$ allows to easily identify which nodes with connection $\epsilon$ we could build next. Taking the submatrices for all values of $\epsilon$, we obtain the target action distribution by setting the entries corresponding to already performed actions at 0, and normalizing the resulting tensor. Fig. 3 illustrates this action tensor building procedure.

**Data Augmentation** The first data augmentation we employ consists of using the same AIG for both targets $T_\star$ and $\neg T_\star$. This is valid because we can generate one target or the other by connecting the final node $\wedge_N$ with the output $O$ using a regular or an inverter edge. Our second data augmentation leverages the fact that any order of truth table rows is valid, provided that the same order is used for all the nodes. Since our ShortCircuit's inputs are truth tables, it is desirable for the model to be invariant to row permutations. Formally, the model should generate the same next-action prediction whether it receives the truth tables $T_1, \ldots, T_N$ and $T_\star$, where $T_i = (t_i^{(1)}, \ldots, t_i^{(2^n)}) \in \{0, 1\}^{2^n}$, or

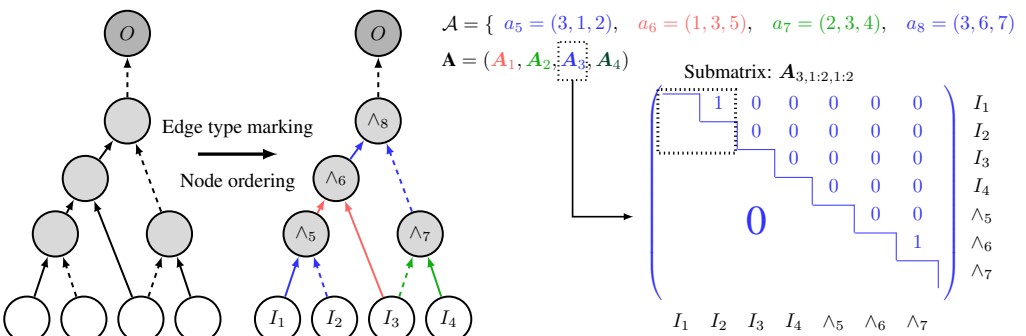

Figure 3: We start data pre-processing by sorting the AIG nodes in topological order. Then, we identify the action types $\epsilon \in \{1, 2, 3, 4\}$ based on the edges. Next, we build the sequence of actions $\mathcal{A}$ and generate the global action tensor $\boldsymbol{A} = (\boldsymbol{A}_1, \boldsymbol{A}_2, \boldsymbol{A}_3, \boldsymbol{A}_4)$. We highlight the structure of $\boldsymbol{A}_3$, which contains a 1 at entries $(1, 2)$ and $(6, 7)$ for the generation of $\wedge_5$ and $\wedge_8$ (actions $a_5$ and $a_8$).

when it gets the permuted truth tables $\sigma(T_1), \ldots, \sigma(T_N), \sigma(T_\star)$ where $\sigma$ is a permutation in $\mathbb{S}_{2^n}$ and $\sigma(T_i) = \left( t_i^{(\sigma(1))}, \ldots, t_i^{(\sigma(2^n))} \right)$. As structurally encoding this invariance into our policy architecture would be computationally too expensive, we apply random permutations to the inputs of our model during the training, which does not impact the other metadata introduced in the previous section.

**Pre-Training Flow**  With the prepared augmented data, we can proceed to train our policy module to match the ground-truth next-action distributions of our training set. For training loss, we experimented with KL divergence and cross-entropy, both of which measure the distance between two probability distributions. In practice, KL divergence loss yielded better results. Besides, the backbone of our model being a transformer, we implement a custom masking strategy during training. to maintain causality in the auto-regressive generation process. Since the primary inputs and the target truth table are available from the start, and as there is no causality for their existence, we allow full attention for their embeddings, and only apply a causal mask for the rest of the nodes.

## 5.2 Fine-Tuning

Fine-tuning aims to align the value and policy module to operate effectively together. Unlike the policy module, we cannot properly initialize the value module during pre-training as the generated dataset only contains successful examples, which would mislead the value module to consider that all states are "good". Skipping pre-training, though, would lead to a random exploration of the vast search space of truth tables, which would likely result in encountering only "bad" states, preventing the model from learning what a "good" state is. Therefore, the most viable option to train our value module is through experience, by performing searches with a pre-trained policy module. We utilize AlphaZero (see Appendix D) as the orchestration framework to refine the policy and value modules.

**Fine-Tuning Flow**  Generating trajectories for millions of truth tables is computationally challenging. Thus, to best exploit our resources, our fine-tuning regimen consists of data collection and model training processes. These data collectors generate trajectories and add their findings to a fixed length replay buffer. Under the hood, the data collectors store the metadata, including truth tables and discovered reward $Q(s)$, of the MCTS root node for each step in the trajectory in the replay buffer. Successful trajectories receive a reward of 1, while failed ones receive $-\min\left(h_d(T_N, T_\star), h_d(T_N, \neg T_\star)\right)$, where $h_d$ is the normalized Hamming distance and $T_N$ is the last generated truth table. The trainer process randomly samples data from the replay buffer and uses the truth tables as input for the model. Since the value module aims to predict the Q-value of a state, the goal is to minimize the mean squared error (MSE) between the predicted value and the retrieved $Q(s)$. The target distribution for the policy module is the normalized number of visitations $N(s,a)/\sum_a N(s,a)$. Similar to pre-training, we minimize the KL-divergence between the policy module output and the target probability distribution.

# 6 EXPERIMENTAL EVALUATION

We introduce the implementation details such as model and search hyperparameters, datasets and baseline methods in section 6.1 and 6.2, and we compare the effectiveness of ShortCircuit against several baselines in 6.3. Finally, section 6.4 presents a study on the impact of number of simulations. We provide further details and discussion about the implementation, training, and scalability of our method in Appendices C, D, and E.

## 6.1 EXPERIMENTAL SETUP

We train and evaluate ShortCircuit with 51.6 million parameters on 8-input truth tables randomly extracted from the EPFL benchmark, as we describe in section 5.1. We specifically choose to test on these circuits, since they correspond to real-world Boolean functions that have more practical interest than uniformly random truth-tables. In total, we extract 1.8 million AIGs with an average number of AND-nodes of 10.08, as we detail in Appendix E.

## 6.2 BASELINES

We derive each truth table in our test set from the primary output of an extracted cut. Since those cuts are AIGs realizing those truth tables, we use them as a baseline, denoted as `Cut`. Additionally, we compare ShortCircuit against the state-of-the-art open-source logic synthesis tool `ABC`. This library applies a series of Boolean algebra transformations to generate an AIG from a truth table. Moreover, we leverage a popular logic optimization flow, `resyn2`, that applies multiple operators to optimize the AIGs by `ABC` and denote it as `ABC+resyn2`. Finally, we compare ShortCircuit against the Boolformer. This learned method produces an optimized Boolean expression given a truth table, and we convert this expression into an AIG without introducing any logic redundancy.

## 6.3 GENERATION QUALITY & RUNTIME EXPERIMENTS

We evaluate ShortCircuit against our baselines on 500 truth tables associated with randomly sampled AIGs from the EPFL benchmarks. We allow ShortCircuit to attempt to generate a circuit with up to 30 AND-nodes. ShortCircuit performs 8 MCTS simulations and generates up to 20 AND-nodes in each simulation, before performing an action. The success rate of ShortCircuit on this test set is 98%.

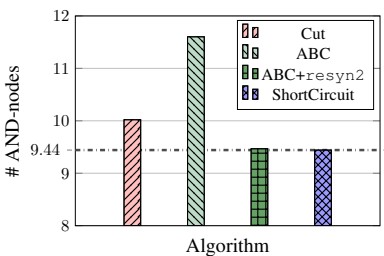
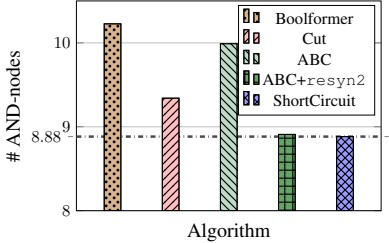

(a) ShortCircuit vs. traditional methods    (b) ShortCircuit vs. traditional and learned methods

Figure 4: Average number of AND-nodes for the successfully generated AIGs across several baselines.

Fig. 4a compares the average size of the successfully generated circuits compared to `Cut`, `ABC`, and `ABC+resyn2`. ShortCircuit directly generates compact AIGs with 9.44 AND-nodes on average. These AIGs are significantly smaller than the ones from `Cut` and `ABC`, respectively achieving a size reduction of 5.77% and 18.62%, and marginally smaller AIGs than `ABC+resyn2` by 0.26%.

Fig. 4b compares the AIG sizes against the same baselines but also against Boolformer. Boolformer successfully generated 85% of the given truth tables, so we report results only on the truth tables successfully generated by both learned methods. ShortCircuit still maintains its good performance on this test subset while Boolformer produces less compact AIGs and frequently fails to synthesize complex circuits requiring more AND nodes. ShortCircuit's AIGs are 13.14% smaller.

The runtime of ShortCircuit for a single rollout with 30 steps is $0.01s$. Due to limitations in our MCTS implementation, the runtime for 8 simulations per step is $1.8s$, but an optimized version would be comparable by performing batched simulations. When we compare the greedy rollout against `ABC` and `ABC+resyn2`, which have a runtime of $0.31s$ and $0.32s$, ShortCircuit achieves a respective speedup of $31\times$ and $32\times$. Finally, Boolformer's runtime is $0.78s$, making ShortCircuit $78\times$ faster.

### 6.4 IMPACT OF NUMBER OF SIMULATIONS

To better understand the role of MCTS simulations in the performance of ShortCircuit, we investigate their impact on the success rate, the circuit size, and the execution time on the same test set of 500 truth tables as in section 6.3. Fig. 5 illustrates how the success rate evolves as the number of MCTS simulations increases from 1 to 256. For clarity, we append an integer $i$ next to our method's name (ShortCircuit[$i$]) to indicate the number of simulations we perform.

When using only 1 simulation, ShortCircuit[1] performs a greedy search, where the policy selects the most likely action, $a = \operatorname{argmax}_{a \in \mathcal{A}} P(s, a; \theta)$. This strategy yields a lower success rate of $92.2\%$ albeit still good, but benefits from a very short generation time of $0.01s$. On the other end, ShortCircuit[256] achieves a significantly higher success rate of $98.6\%$, albeit with a much longer running time of $106s$. Increasing the number of simulations enables the model to explore a larger – but still limited – portion of the solution space, resulting in higher success rates and the discovery of more compact graphs. Fig. 6 highlights this trade-off by revealing a Pareto front, suggesting that we can adjust the number of MCTS simulations to achieve the desired balance between success rate, design quality, and running time. Moreover, Fig. 6 confirms that, with more engineering efforts, the runtime for different numbers of simulations would be much closer to ShortCircuit[1], since the runtime scales sublinearly with respect to the number of simulations as already visited states do not need to be recomputed and are retrieved from cache.

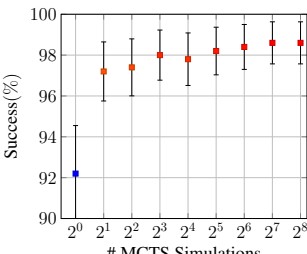 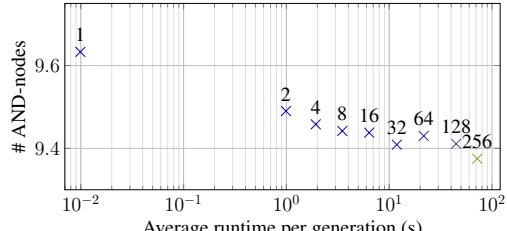

Figure 5: ShortCircuit's success rate vs. number of MCTS simulations per action.

Figure 6: Average #AND-nodes per AIG vs. generation time for ShortCircuit with varying MCTS simulations marked on top of each data point.

## 7 CONCLUSION

In this paper, we introduced ShortCircuit, a novel transformer-based architecture for generating AIGs from a target truth table. Our approach combines a structurally aware transformer model with an AlphaZero-inspired policy variant, enabling efficient navigation through the doubly exponential state space associated with truth tables. In our experiments, we demonstrated the effectiveness of ShortCircuit in producing high-quality AIGs that are significantly smaller than those generated by one of the state-of-the-art logic synthesis tools `ABC` and the trained Boolformer. Specifically, our method achieved a relative size reduction of $18.62\%$, and $15.13\%$ respectively.

This work contributes to the expanding field of ML applications in chip design, showcasing the potential of deep learning to revitalize the field with new perspectives. We demonstrated that it is possible to generate high-quality AIGs from truth tables, paving the way for future research in this area. Future work will focus on extending ShortCircuit to handle AIGs with multiple outputs, integrating our approach with existing logic synthesis tools, and exploring its application in industrial settings. Finally, our goal is to enable the creation of efficient, scalable, and innovative computing systems, and we believe that ShortCircuit is an important step towards realizing this vision.

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

## A    NOTATIONS

Table 3 contains a condensed summary of the notation introduced throughout the paper.

| Symbol | Meaning |
|---|---|
| $n$ | Number of inputs in the AIG |
| $I_j$ | Input node $j$ in the AIG |
| $O$ | Output node for the AIG |
| $\wedge_i$ | AND-node $i$ for the AIG |
| $T_\star$ | Target truth table |
| $s$ | State |
| $a$ | Action |
| $\mathcal{T}$ | Set of truth tables in the AIG |
| $\mathcal{A}$ | Set of actions |
| $N$ | Current number of nodes in the AIG |
| $N_{\max}$ | Max number of nodes allowed in the AIG ShortCircuit generates |
| $\epsilon$ | Building type of an AND-node with respect to its two fanins (Table 2) |
| $\mathbf{A}$ | Sparse 3-dimensional tensor accumulating all the target actions |
| $\mathbb{S}_k$ | Set of permutations of $\{1..k\}$ |
| $\sigma(\cdot)$ | Random row permutation function |
| $Q(s, a)$ | Discovered Q-value for a state $s$ and action $a$ |
| $Q(s, a; \theta)$ | Predicted expected Q-value for a state $s$ and action $a$ |
| $P(s, a; \theta)$ | Predicted action probability distribution |
| $N(s, a)$ | Visit count of action $a$ and state $s$ |
| $b, c$ | Parameter balancing exploration and exploitation in PUCT |

Table 3: List of symbols and notations used in the paper.

## B    ADDITIONAL RELATED WORKS

### B.1    HEURISTICS FOR AIG GENERATION AND OPTIMIZATION

As the inference of CNFs using exact SAT solvers often lead to exponentially large expressions, various heuristics such as Karnaugh maps (Karnaugh, 1953), or Quine-McCluskey methods (Quine, 1952; 1955; McCluskey Jr., 1956)), and algorithms (Rudell and Sangiovanni-Vincentelli, 1987) have been designed to obtain more compact expressions or circuits. Further efforts accompanying the rise in chip demand led to the development of widely used logic synthesis libraries that implement equivalence-preserving Boolean network operators. The open-source library `ABC` (Mishchenko et al., 2007)) notably comprises dozens of logic graph operators aiming at reducing a network size or depth (Mishchenko et al., 2011; Mishchenko and Brayton, 2006). Interestingly, some important operators such as `resub` or `rewrite` (Darringer et al., 1981; Mishchenko et al., 2006) acts on the subject graph through a series of local modifications involving small single-output AIGs. Besides, applying a single operator on a logic network is suboptimal compared to applying several operators sequentially, though finding the best sequence is also a hard problem (Riener et al., 2019).

### B.2    MACHINE LEARNING FOR LOGIC SYNTHESIS

Many ML approaches have been explored to tackle the operator flow optimization progress. Some stateless optimization methods, such as Bayesian optimization (Grosnit et al., 2022; Feng et al., 2022), search for the best flow without considering the subject graph specificities. Alternatively, state-based methods formulate the operator sequence optimization as a Reinforcement learning problem, and train policies on selected features of the logic network. While some works use high-level statistics of

the subject graph (e.g., its number of nodes) (Hosny et al., 2020; Zhou and Anderson, 2023; Qian et al., 2024), others rely on tailored graph convolutional networks (GCN) to extract richer features at the cost of a longer training time (Haaswijk et al., 2018; Peruvemba et al., 2021; Zhu et al., 2020; Basak Chowdhury et al., 2023). Similarly, standard deep network architectures, such as CNNs (Yu et al., 2018), LSTMs (Yu and Zhou, 2020), or GCNs (Wu et al., 2022) have been trained to predict the quality of a logic synthesis flow in a supervised way. Contrary to these works, we target AIG generation itself and not operator optimization.

## C DATA COLLECTION

### C.1 EPFL BENCHMARKS

Tables 4 and 5, contain detailed information about the arithmetic and random control circuits in the EPFL benchmarks (Amarú et al., 2015), respectively. The circuits have been mapped from behavioral descriptions into logic gates and are intentionally suboptimal for scientific purposes. Arithmetic circuits, as their name hints, are combinatorial AIGs representing an arithmetic operation such as square root, logarithm, etc., while the set of random control circuits consists of controller circuits.

| Circuit Name | # Inputs | # Outputs | # AND-nodes | Levels |
|---|---|---|---|---|
| Adder | 256 | 129 | 1020 | 255 |
| Barrel Shifter | 135 | 128 | 3336 | 12 |
| Divisor | 128 | 128 | 44762 | 4470 |
| Hypotenuse | 256 | 128 | 214335 | 24801 |
| Log2 | 32 | 32 | 32060 | 444 |
| Max | 512 | 130 | 2865 | 287 |
| Multiplier | 128 | 128 | 27062 | 274 |
| Sine | 24 | 25 | 5416 | 225 |
| Square-root | 128 | 64 | 24618 | 5058 |
| Square | 64 | 128 | 18484 | 250 |
| **Average:** | 166 | 102 | 37396 | 3608 |

Table 4: Arithmetic circuits in the EPFL benchmark suite and their statistics

| Circuit Name | # Inputs | # Outputs | # AND-nodes | Levels |
|---|---|---|---|---|
| Round-Robin Arbiter | 256 | 129 | 11839 | 87 |
| Alu Control Unit | 7 | 26 | 174 | 10 |
| Coding-Cavlc | 19 | 11 | 693 | 16 |
| Decoder | 8 | 128 | 304 | 3 |
| i2c Controller | 147 | 142 | 1342 | 20 |
| Int to Float Converter | 11 | 7 | 260 | 16 |
| Memory Controller | 1204 | 1231 | 46836 | 114 |
| Priority Encoder | 128 | 8 | 978 | 250 |
| Lookahead XY Router | 60 | 30 | 257 | 54 |
| Voter | 1001 | 1 | 13758 | 70 |
| **Average:** | 284 | 171 | 7644 | 64 |

Table 5: Random/Control circuits in the EPFL benchmark suite and their statistics

### C.2 CUT EXTRACTION

To extract a cut with a target number of inputs $n$ from a circuit and a given node, we set the node as the root node and initialize the leaf set with the two parents of the root node. To extract a cut from a given node with a target number of inputs $n$, we designate the given node as the root and of the cut and its two parents as the initial leaf set. To extract a cut with $n$ inputs from a circuit at a given node, we set the node as the root and initialize the leaf set with its two parents. Then, we iteratively remove a random node from the leaf set and add its parents to the leaf set while maintaining the leaf property.

This process continues until the leaf set contains $n$ nodes. Finally, we create an AIG from the visited nodes within the cut, where we mark the leaf set as the inputs and the root node as the output. We provide the outline of the procedure in Algorithm 1.

---

**Algorithm 1** Cut Extraction

---

**Require:** Root node: $r$, number of cut inputs: $n$
1: leaf_set = {left_parent($r$), right_parent($r$)}
2: node_set = {$r$}
3: **while** size(leaf_set) < $n$ **do**
4:  node = leaf_set.random_pop()
5:  node_set.insert(node)
6:  leaf_set.insert(left_parent(node)) & leaf_set.insert(right_parent(node))
7:  Ensure leaf property in leaf_set
8: Construct AIG from leaf_set and node_set

---

We can modify this algorithm to extract additional cuts per node by repeating the process until we find a cut with $n - 1$ leaf nodes. For this cut, instead of randomly expanding a leaf node, we create $n - 1$ copies of the cut and expand each leaf node individually, storing the resulting cuts. In practice, we actually employ Algorithm 2 to extract AIGs from the EPFL circuits. Although Algorithm 1 conveys the core idea of AIG extraction, the following algorithm is more effective from an engineering standpoint, as it allows for extracting more cuts from the same node.

---

**Algorithm 2** Multi-Cut Extraction

---

**Require:** Root node: $r$, Number of cut inputs: $n$
1: leaf_set = {left_parent($r$), right_parent($r$)}
2: node_set = {$r$}
3: **while** size(leaf_set) < $n - 1$ **do**
4:  node = leaf_set.random_pop()
5:  Cut_Expansion(node, leaf_set, node_set)
6: **for** leaf in leaf_set **do**
7:  copy_leaf_set = leaf_set.copy() & copy_node_set = node_set.copy()
8:  copy_leaf_set.delete(leaf)
9:  Cut_Expansion(leaf, copy_leaf_set, copy_node_set)
10: Construct AIG from copy_leaf_set and copy_node_set

---

**Algorithm 3** Cut Expansion

---

**Require:** Node to expand: node, Current leaf nodes: leaf_set, Current nodes in cut: node_set
1: node_set.insert(node)
2: leaf_set.insert(left_parent(node)) & leaf_set.insert(right_parent(node))
3: preserve_leaf_property(leaf_set, node_set)

---

**Algorithm 4** Preserve Leaf Property

---

**Require:** Current leaf nodes: leaf_set, Current nodes in cut: node_set
1: **for** leaf in leaf_set **do**
2:  **if** left_parent(leaf) in leaf_set **then**
3:   leaf_set.delete(leaf) & node_set.insert(leaf)
4:   leaf_set.insert(right_parent(node))
5:  **else if** right_parent(leaf) in leaf_set **then**
6:   leaf_set.delete(leaf) & node_set.insert(leaf)
7:   leaf_set.insert(left_parent(node))

---

The revised algorithm takes two inputs: $n$, the desired number of input nodes, and the root node. It initializes the leaf set by adding the parents of the root node. It then iteratively removes a random node from the leaf set and expands it using Algorithm 3. This process continues until the cut contains

$n - 1$ leaf nodes, at which point we create $n - 1$ copies of the current cut state and expand each leaf node to generate $n$ unique cuts. For brevity, we omitted the details of ensuring the leaf property after node expansion in the main text, which is addressed in Algorithm 4. This algorithm ensures that no node in the leaf set has a parent also in this set, which would violate the leaf property. If a parent is already in the leaf set, we can remove the node from the set and add the other parent that is not yet in the leaf set.

## D  ADDITIONAL METHODOLOGY DETAILS

**ShortCircuit Generation Example**   The visual representation of the model is depicted in Fig. 2, and the edge type indices are shown in Table 2. In the beginning, the graph contains only the input nodes, as shown below. The input for the model will be all the truth tables. Specifically, that includes the truth tables of the inputs $T_1, T_2, T_3$ and the target truth table $T_\star$.

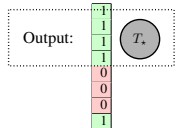

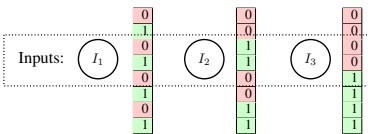

ShortCircuit produces action logits, where taking the arg max results into the action $(1, 1, 2)$. The indices of the action that we have to generate the AND node $\wedge_4$ by connecting $(I_1, I_2)$. Given that we know the truth tables of the parents of $\wedge_4$, we can calculate its truth table. Since the truth table of $\wedge_4$ does not match $T_\star$ or $\neg T_\star$, we need to continue the generation. The model will receive again as input all the current truth tables, which now includes the one associated with $\wedge_4$.

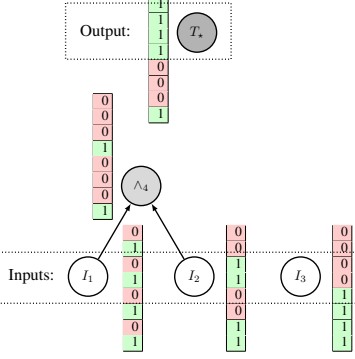

This time, the proposed action from ShortCircuit is $(3, 3, 4)$ which generates the new AND gate $\wedge_5$ by connecting $(I_3, \neg \wedge_4)$. The new truth table associated with $\wedge_5$ matches with $\neg T_\star$. Meaning we do not need to generate any more nodes.

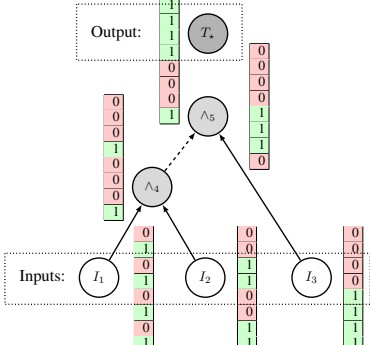

Thus, the last step is to connect $\wedge_5$ with a negated edge to the output and finish the generation.

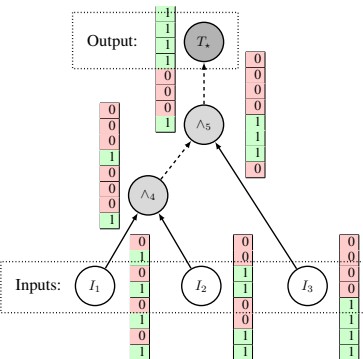

**Scalability & Multiple-Outputs**  The architecture of ShortCircuit can technically support any size of truth tables. The input truth table size is fixed but user-defined and can be set to any $2^n$ size where $n$ is the number of variables. The model down-projects these truth tables to a fixed embedding size in continuous space, which improves computational efficiency. In our current implementation, we project truth tables with 256 bits to float embeddings of size 256.

Our model architecture also supports multiple output nodes. Users can include several target truth tables as input to ShortCircuit, with the model injecting the relevant positional embeddings for each. A simpler alternative approach for handling multiple outputs is sequential generation: first generate an AIG for the initial target truth table, then replace it with the second target table and continue generation. This process can be repeated until all target truth tables are satisfied.

**AlphaZero**  AlphaZero has demonstrated remarkable success in board games with enormous state spaces, such as chess ($10^{44}$) and Go ($10^{170}$). Since truth tables features similar state space problem, we adapt AlphaZero's effective search and pruning capabilities to navigate AIG generation. By combining a policy module to propose actions and a value module to evaluate state viability, AlphaZero strikes a balance between exploitation and exploration. We adapt and modify the selection strategy, predictor upper confidence bound applied to trees (PUCT) used by AlphaZero, as follows:

$$\text{PUCT}(s,a) = Q(s,a) + b\,Q(s,a;\theta) + c\,P(s,a;\theta)\frac{\sqrt{\sum_a N(s,a)}}{N(s,a)+1}$$

where, $Q(s,a)$ represents the propagated discounted discovered reward, while $Q(s,a;\theta)$ represents the predicted expected Q-value, $P(s,a;\theta)$ is the policy module's probability distribution, $N(s,a)$ tracks state visitations, and $b$ and $c$ are parameters balancing exploration and exploitation. Computing $Q(s_t,a;\theta)$ for every action is too expensive, so we initialize $Q(s_t,a) = Q(s_t,a;\theta) = 0$, perform the action that maximizes $\text{PUCT}(s_t,a)$, and only compute the value of the state $Q(s_{t+1})$ once we visit it. The term $Q(s,a) + bQ(s,a;\theta)$ represents the exploitation in PUCT, as if during search our method discovers a "good" state or a terminal state, we exploit it and focus the search locally to discover more compact designs. The term $P(s,a;\theta)$ suggests actions to perform, but the term $\sqrt{\sum_a N(s,a)}/(N(s,a)+1)$ promotes exploration.

AlphaZero stores intermediate results and metadata, such as $Q(s,a)$, $Q(s,a;\theta)$, $P(s,a;\theta)$, and $N(s,a)$, in the nodes visited during MCTS. These nodes are associated with states and form a tree, where edges indicate the actions performed to reach each node-state pair. When simulation starts, we mark the initial state as the root node, compute the action distribution, and inject Dirichlet noise. During simulation, AlphaZero follows PUCT to choose actions and continues until meeting one of the three following stopping conditions: encountering a state $s$ that is not expanded, reaching a maximum number of steps, or arriving at a terminal state. If the state is not expanded, we need to compute $Q(s;\theta)$ and $P(s,a;\theta)$ for that state and we back-propagate $Q(s;\theta)$ to the previous MCTS nodes, and increment $N(s,a)$. Once we complete the given number of simulations, AlphaZero applies the most visited action, $\arg\max_{a \in \mathcal{A}} N(s,a)$. In our case, we rather follow the observed discounted reward $\arg\max_{a \in \mathcal{A}} Q(s,a)$ as we find the visitation count signal too noisy given our simulation budget.

# E    TRAINING PARAMETERS AND IMPLEMENTATION

**Implementation Details**    We implement ShortCircuit with PyTorch (Paszke et al., 2019) and TorchRL (Bou et al., 2023). Our model architecture is as depicted on Fig. 2 and uses transformer blocks following Llama 3 (Meta Llama team, 2024) structure. The input truth tables have a size of 256 since they are dependent on 8-inputs and are projected into embeddings of size 256 in the continue space. The input embedding size for our transformer decoder layers is 256 since they receive those hidden states as input. We use are $H = 4$ and $P = S = 3$ transformer blocks for the different parts with 16 heads and an intermediate embedding size of 4096, summing to 51.6 million parameters.

**Model Details**    During pre-training, We use a cosine annealing with warm restarts learning rate scheduler with a starting learning rate of $1 \times 10^{-3}$ and a batch size of 1024 for 250 epochs. From the extracted data, we extract 500 truth tables for testing, the remaining data is split into 90% for training, 10% for validation. The entire dataset contains about 1.8 million (AIG, truth table) pairs. During training we apply a random permutation to the rows of the truth tables with probability and a negation transform to the target with probability 50%. We pre-train ShortCircuit with a batch size of 1024 for 250 epochs, and finetune the model until it converges. Finally, to speed-up pre-training, we use a distributed dataloader yielding batches of sequences of same length to avoid applying padding and consequently unnecessary computations. During fine-tuning we use a batch size of 128, a replay buffer with capacity of 1M and sync the parameters every 500 training steps.

**Baseline Details**    The specific sequence of commands we use in `ABC` to generate AIGs from truth tables in ABC are

- `ABC`:
  ```
  read_truth -x [truth table]; collapse; sop; strash; write
  [outfile].
  ```
- `ABC+resyn2`:
  ```
  read_truth -x [truth table]; collapse; sop; strash; resyn2;
  write [outfile].
  ```

Specifically, `resyn2` consists of the following commands `b; rw; rf; b; rw; rwz; b; rfz; rwz; b`.

