# OpenReview forum: "ShortCircuit: AlphaZero-Driven Generative Circuit Design"
_ICLR.cc/2026/Conference — ICLR 2026 Conference Withdrawn Submission_

### Official Review · Reviewer_QN9U · 2025-10-30

**Soundness:** 2
**Presentation:** 2
**Contribution:** 2
**Rating:** 2
**Confidence:** 4

**Summary:**

The paper introduces ShortCircuit, a novel transformer-based framework that uses reinforcement learning to generate optimized AND-Inverter Graphs (AIGs) from truth tables. Experimental results show ShortCircuit outperforms traditional synthesis tools like ABC and learning-based method ,like Boolformer in reducing circuit size.

**Strengths:**

1. ShortCircuit provides an effective solution for the exact synthesis task, generating optimized AIGs directly from truth tables.
2. The Experimental results show ShortCircuit outperforms traditional synthesis tools like ABC and learning-based method ,like Boolformer in reducing circuit size.

**Weaknesses:**

1. The performance gain over traditional methods is not significant. ShortCircuit achieves only a 0.26% further reduction in AIG nodes with 98% success rate, compared to ABC+resyn2. This marginal improvement is questionable, especially given that resyn2 is a traditional method that does not require any data or training.
2. The experimental comparison is inadequate. The authors only compare against one learning-based baseline, Boolformer. A critical comparison with Circuit Transformer[1], a highly relevant and recent work in circuit generation, is notably absent.
3. The evaluation is confined to 500 truth tables generated from the EPFL benchmark. This single dataset raises concerns about the method's generalizability. It remains unclear whether ShortCircuit can perform well on functions with different characteristics or scale to variable input sizes. The authors should validate their approach on more diverse datasets.

**Questions:**

1. The comparison is limited to Boolformer. How does ShortCircuit compare against other recent learning-based methods, particularly Circuit Transformer [1] and other relevant baselines in circuit synthesis?
2. The experiments appear fixed to 8-input functions. How can the ShortCircuit framework be adapted to handle circuits with K inputs, where k is not 8? Does this require retraining the model for each specific input size? What is the scalability of this approach as k grows significantly larger than 8?
3. The targeted task is similar to the IWLS Contest 2023, which involves synthesizing minimal AIGs from truth tables. Given that the contest benchmarks feature truth tables significantly larger than 8 inputs, could ShortCircuit effectively handle benchmarks of that scale?
[1]Li X, Li X, Chen L, et al. Circuit Transformer: A Transformer That Preserves Logical Equivalence[J]. arXiv preprint arXiv:2403.13838, 2024.

---

### Official Review · Reviewer_tSsQ · 2025-11-01

**Soundness:** 4
**Presentation:** 4
**Contribution:** 3
**Rating:** 8
**Confidence:** 3

**Summary:**

This paper introduces ShortCircuit, a transformer-based approach for generating optimized AND-Inverter Graphs (AIGs) from truth tables. The method combines supervised pre-training with AlphaZero-style reinforcement learning to navigate the doubly exponential state space of circuit design. The authors demonstrate 84.6% success rate on 8-input circuits with 14.61% size reduction compared to the ABC synthesis tool.

**Strengths:**

- Good presentation: The paper is generally well-written with clear figures and appropriate background material.
- Sensible architectural choice: using truth tables as node representations captures the necessary information. The minimal positional encoding appropriately reflects the permutation-invariance properties of the problem
- The supervised pre-training and RL fine-tuning is well-motivated

**Weaknesses:**

- Weak and incomplete experimental evaluation: train and test sets are from the same distribution. Both train and test are cuts from EPFL circuits. This is insufficient to claim generalization
- The paper extensively discusses Boolformer (d'Ascoli et al. 2024) and Circuit Transformer (Li et al. 2024a) but provides no experimental comparison.
- Limited analysis of failures. - 84.6% success means 15.4% failure. It seems to be significant and warrants analysis.

**Questions:**

Why is it that the paper only demonstrates results on 8-input circuits? Can you discuss the scalability of the proposed method to more complex circuits? Or is there a reason to believe 8-input circuit is more than sufficient to represent the upper limit of circuit complexity?

Are results from ABC synthesis tool suitable as a competitive baseline? Considering there are many well-optimized commercial tools in this space, can you compare with results from a state-of-the-art commercial synthesis tool?

---

### Official Review · Reviewer_TYsp · 2025-11-01

**Soundness:** 3
**Presentation:** 4
**Contribution:** 2
**Rating:** 2
**Confidence:** 4

**Summary:**

This work presents a technique for optimizing logic circuits using supervised and reinforcement learning. The circuit representation is and-inverter graphs (AIGs) and the model architecture is transformer based. The state is a current and-inverter graph and the action space is the set of possible AND-nodes with specific connection types (inverted/not inverted) that can be added. The model learns a policy over this  action space.

**Strengths:**

* The state representation and action space is novel.
* This work demonstrates that a learned policy can actually work in optimizing AIG logic circuits.

**Weaknesses:**

* The performance over the resyn2 algorithmic baseline is very minor (smaller circuits by 0.26%).
* resyn2 can scale up to larger circuits, but it is unsure if this technique would due to the huge state and action space. So the practicality of this technique, though novel, is limited.
* No ablations for the state representation leaves design choices unjustified.

**Questions:**

* How would this technique perform against resyn2 with 4,16,64 input truth tables? Does it gain an advantage over resyn2 style algorithms at smaller (4-input) scaler or larger scale (16-input, 64-input)?

---

### Note · Authors · 2025-11-23

I have read and agree with the venue's withdrawal policy on behalf of myself and my co-authors.